# Diagnostic Uncertainty: Teaching Language Models to Describe Open-Ended Uncertainty

**Brian Sui**[*], **Jessy Lin**[*], **Michelle Li**[*], **Anca Dragan, Dan Klein, Jacob Steinhardt**
University of California, Berkeley
{brian.sui, jessy_lin, michelle_li}@berkeley.edu

## Abstract

Language models (LMs) often hallucinate. While uncertainty measures like calibration scores provide coarse measures of model uncertainty (e.g. "This proof is 40% likely to be correct"), ideally a model could tell us *what* it's uncertain about, such as "I don't know how to find the length of side AB," enabling people to understand exactly where to trust a model response. We propose **diagnostic uncertainty**: open-ended descriptions of uncertainty that are grounded in model behavior. Our key idea is that a model can be said to be uncertain about $X$ (e.g., "how to find the length of side AB") if its responses significantly improve after being told $X$, and $X$ is earliest in its reasoning process. We implement a method to bootstrap models' ability to generate these diagnostic uncertainty descriptions by iteratively training on sampled descriptions that satisfy these criteria. To evaluate whether diagnostic descriptions are meaningful, we provide the model with the information it claims to be uncertain about and measure whether its performance improves. Compared to the descriptions generated by prompting alone, resolving diagnostic uncertainty descriptions leads to 8% higher accuracy and 20% more reduction in entropy of the answer distribution, supporting the hypothesis that diagnostic uncertainty is more faithful to the model's underlying uncertainty. The main contribution of our work is a framework for operationalizing open-ended uncertainty in LMs, enabling richer ways for people to understand LM behavior beyond raw probabilities.

## 1 Introduction

Language models (LMs) are becoming increasingly popular tools and real-world assistants, so it is crucial that users know when to trust their outputs. There are a myriad of possible sources of uncertainty, whether they are facts about the world that are constantly being updated or simply weren't in the training data, technical concepts that are tricky to implement perfectly, or ambiguity in a problem specification. Even when augmented to use external sources or tools, there will always be things LMs do not know. Whenever this is true, we would like LMs to express their uncertainty, but unfortunately, language models often hallucinate or confidently output false statements, so they must be trained to express uncertainty.

One approach is to train models to qualify their answers with confidences like "I'm 70% confident that...". While linguistic confidences provide a useful score of how much a model's answer can be trusted, probabilities are not a natural way for end users to understand uncertainty, especially in long-form generations or open-ended domains such as coding where it would be impractical to quantify the confidence at every step.

Ideally, we want models to express not only *how much* they're uncertain in a response, but *what* or *why* they're uncertain, e.g. "I'm not sure how to implement the boundary condition for the algorithm," or "I don't remember exactly whether the date was 1890 or 1990." Practically, this kind of **open-ended uncertainty** would allow end users to understand exactly when to trust model responses, enabling models and humans to collaborate to answer increasingly difficult questions where neither has the answer. This kind of uncertainty is also reminiscent of the introspective process that humans engage in when learning or solving problems: thinking about one's own thinking, or *metacognition* (Flavell, 1976), is an important feature of reasoning. In contrast to verbalized calibration scores, open-ended uncertainty can enable richer ways to understand and improve models.

**Question**: Two sides of an isosceles triangle are 10 inches and 20 inches. If the shortest side of a similar triangle is 50 inches, what is the perimeter of the larger triangle?

**Model**: To solve the problem, we first identify the sides of the original isosceles triangle. The triangle has two sides of length 10 inches (the equal sides) and one side of length 20 inches.

...

## Answer: 200  ❌ **Incorrect: True answer "250"**

**Calibration-Based Uncertainty**

```
Confidence: 60% / Medium
```

**Diagnostic Uncertainty**

```
One step where I was most uncertain was during
the identification of which configuration of
the isosceles triangle with sides 10 inches and
20 inches was valid and applicable for the
problem. How can I definitively confirm which
configuration of the triangle (10-10-20 or
20-20-10) is valid based on the triangle
inequality theorem?
```

Figure 1: Our goal is to enable people to understand what an LM is uncertain about, beyond calibration scores that only communicate the likelihood a response is wrong. In contrast, we propose diagnostic uncertainty: a way to train LMs to output open-ended uncertainty descriptions—identifying ambiguous questions, knowledge gaps, and reasoning errors in a unified way. The figure above shows a diagnostic uncertainty generated by our trained model on a MATH example, where it identifies that it may have chosen the configuration of the triangle incorrectly.

How can we extract models' open-ended uncertainties? We can't simply ask models what they're uncertain about and trust that their responses are faithful to their true uncertainty because the standard training methods for language models do not incentivize faithfulness by default. We could try to train models to report their uncertainties, via supervised fine-tuning, but it is unclear how to get labels for what models are actually uncertain about. And certainly, we do not want models to imitate what human annotators are uncertain about, as this may be different from what *the model* is uncertain about.

In this work, we propose **diagnostic uncertainty**: a way to operationalize open-ended uncertainty grounded in model behavior. Our key idea is that a model can be said to be uncertain about $X$ if knowing $X$ would improve its responses. For example, a model should only say it's uncertain about $X =$ "how to find the length of side AB" if its current solution is incorrect and it is able to correct it after being told how to find side AB. Additionally, we want descriptions of the *root* uncertainty: a model should not say it's uncertain about $X = A + B$ if knowing $A$ alone would be sufficient to correct its solution.

We introduce a method that searches for diagnostic uncertainties via iterative filtering and fine-tuning. We sample an initial set of possible uncertainties from the model, filter for the ones that, when resolved, improve model performance, and fine-tune the model to produce those filtered uncertainties. We iterate this process to bootstrap the model's ability to express its own uncertainty in a self-supervised way.

Our method works with any sort of uncertainty so long as there is a way to measure task performance, whether this is a ground truth answer, solution, or reward model for an open-ended problem.

We test our method on Hendrycks MATH (math reasoning; Hendrycks et al. (2021)), finetuning `gpt-4o-mini` to produce diagnostic uncertainty descriptions. To evaluate whether uncertainty descriptions are better than those generated with prompting alone, we resolve the model's uncertainty by telling it what it doesn't know (using another model with access to the solution), and measure: (1) does the entropy of the answer distribution decrease? (2) does the accuracy increase? Compared to prompting the model to report its uncertainty, diagnostic uncertainty decreases the entropy of the answer distribution by 20% more and accuracy increases by 8% relative to prompting, supporting our hypothesis that diagnostic uncertainty is more faithful to the model's underlying uncertainty. Qualitatively, we show that diagnostic uncertainty indeed enables a richer way to diagnose model errors: we identify hallucinated facts, incorrect reasoning steps, and ambiguities in the dataset question in a unified way.

## 2  RELATED WORK

**Uncertainty Estimation for LMs.** Existing work on uncertainty in LMs has largely focused on calibration of short form answers using token probabilities, ensembling across multiple samples, or verbalized confidences that directly prompt the LM to output its uncertainty in words (Mielke et al., 2022; Kadavath et al., 2022; Lin et al., 2022; Farquhar et al., 2024; Xiong et al., 2024). LMs have been found to have internal representations of their errors (Burns et al., 2022; Orgad et al., 2024;

Problem: What is the greatest possible number of digits in the product of a 4-digit whole number and a 3-digit whole number?

Figure 2: **Conceptual Framework.** To formalize which uncertainty descriptions we want the model to output, we can think of the process of solving a problem as a graph, where each node in the graph is a decision point or computational step. In this example, to determine the number of digits in the product, one node in the graph is to determine the solution approach (sum logarithms, multiply the largest 3-digit and 4-digit numbers, etc.); once an approach is selected, another node is to compute a particular value for the solution. Uncertainty or errors early on in solving a problem affect downstream nodes: if the model can't reliably determine $log10(9999)$, it also can't determine $log10(9999) + log10(999)$. In our framework, we would like the model to describe the first ("root") node where intervening on that node makes the model's solutions significantly more correct ("critical"). Note that the graph is simply a conceptual tool to motivate what open-ended descriptions we would like to output—it is not explicitly generated by our method.

Simhi et al., 2024; Ferrando et al., 2024), and they are surprisingly calibrated even with prompting alone (Tian et al., 2023). However, these outputs may be influenced by spurious correlations to expressions of uncertainty in pretraining data (Zhou et al., 2023). Recent works extend this to the long-form generation case by decomposing long generations into atomic claims (Band et al., 2024; Jiang et al., 2024), or aggregating across many samples (Manakul et al., 2023). We build on prior techniques such as sample aggregation, but aim to produce natural language descriptions of uncertainty for long-form outputs, rather than confidence scores.

Another body of work studies how to enable LMs to identify uncertainty about user queries rather than internal knowledge, training models to generate clarification questions when queries are ambiguous (Rao & Daumé III, 2018; Andukuri et al., 2024; Chi et al., 2024). Rather than handling query ambiguity separately, we propose a unified way to identify when models are uncertain about the question, specific facts, or reasoning steps.

**Reasoning and Metacognition.** Metacognition, or awareness of one's own thought processes, is an important part of reasoning and learning in humans (Flavell, 1976). Some work studies metacognitive abilities in LMs, e.g. to determine which skills they use on a task (Didolkar et al., 2024), predict what they will output (Binder et al., 2024), or why they make classification decisions (Sherburn et al., 2024); we focus on improving LMs' ability to express their own uncertainty. While our work does not test whether whether our outputs represent true introspection or metacognition in LMs, our framework provides one way to ground these ideas in concrete model behavior.

Recent reasoning models output reflections of their own thinking when they are trained to output chains of thought that reach the correct answer (Zelikman et al., 2022; OpenAI, 2024; DeepSeek-AI et al., 2025). Some of these chains of thought emergently express and reason through uncertainty in the course of solving a problem. However, it is unclear whether the uncertainty described in these chains of thought are faithful to the model's actual uncertainty, or just serve to lead the model to the right answer for uninterpretable reasons (Turpin et al., 2023; Lanham et al., 2023). Our work trains these open-ended uncertainty descriptions to be interpretable and grounded in model behavior. Our goal is not to generate descriptions that make model outputs more correct, but to use descriptions to inform human users where they can trust a model's response (Zhou et al., 2024).

**Model Interpretability.** Our method can be viewed as a tool for model interpretability, searching over natural language descriptions that describe knowledge a model is missing when answering a question. Current methods use natural language descriptions as a rich way to label neurons (Hernandez et al., 2022; Bills et al., 2023), elicit harmful behaviors (Li et al., 2024), or identify error patterns in model outputs (Zhong et al., 2023). Like our method, these approaches often leverage LMs themselves to enable the generation and validation of richer, open-ended descriptions (Singh et al., 2024).

## 3 FORMULATION

Our goal is to generate open-ended natural language descriptions of model uncertainty on a particular task, which can include:

- which step of a problem is likely to be incorrect: "I'm not sure what 1578 * 979 is"
- "why" responses might be incorrect: "I'm unsure about how to incorporate the boundary condition"
- what the uncertainty is between: "I'm not sure whether the date is 1980 or 1990"
- ambiguity: "I'm not sure whether you're asking for the name of the chemical, or the name of its solvent"

While LMs can be prompted to superficially generate these descriptions zero-shot, they are not useful unless they meaningfully represent the model's uncertainty or some property of its responses. In this section, we describe **diagnostic uncertainty**, one way of operationalizing natural language descriptions of model uncertainty.

### 3.1 CONCEPTUAL FRAMEWORK

To motivate what open-ended uncertainty descriptions are meaningful, we introduce an intuitive conceptual framework based on computational graphs. Consider the process of solving a problem $p$ as traversing a directed graph $G = (V, E)$ where the final node is the model's answer $a$ and each intermediate node $v \in V$ represents a decision point or computational step. These nodes can represent planning decisions like "decide which algorithm to apply", interpretation steps like "translate 'twice the difference' into an equation", or concrete claims and computations like "determine the square footage of La Norte Mall." We depict this in Figure 2.

One can have a belief distribution about the *value* of each node, i.e., the correct output of that computation. In this view, models not only have a distribution over *answers* (calibration-based uncertainty; uncertainty about the value of the final answer node $a$), but also nodes that represent higher-level abstract reasoning steps. This allows us to express the search space of potential uncertainty descriptions: all possible nodes, in all possible solution graphs.

Importantly, each node's value influences downstream values and the structure of what computations to do later in the graph. If we enumerated all the possible decision points, nodes that the model is highly uncertain about would have large branching factors in structure and values downstream graph. If a model is uncertain about how to interpret the question "What is twice the difference between 12 and 5?" it will also be uncertain about how to write down the equation, the value of the evaluated expression, and the final answer.

Instantiating the space of graphs explicitly would be intractable, but this framework helps us characterize the description we'd like models to output. Intuitively, the first property we want is to find nodes where uncertainty at that node explains the uncertainty in the model's final response. If we intervened by revealing the value of node $v$, how much does the model's accuracy at the final output improve? This rules out nodes where the model is certain and nodes with "irrelevant" uncertainty (e.g., about surface forms)—both of these have no causal effect on improving the final response.

Expressing uncertainty about the final answer node trivially satisfies the first property, but these descriptions are not as informative. Ideally, we should incentivize models to output precise uncertainty, or the most upstream node that accounts for the uncertainty in the response—"I'm unsure how to interpret the question," rather than "I'm unsure about the final answer."

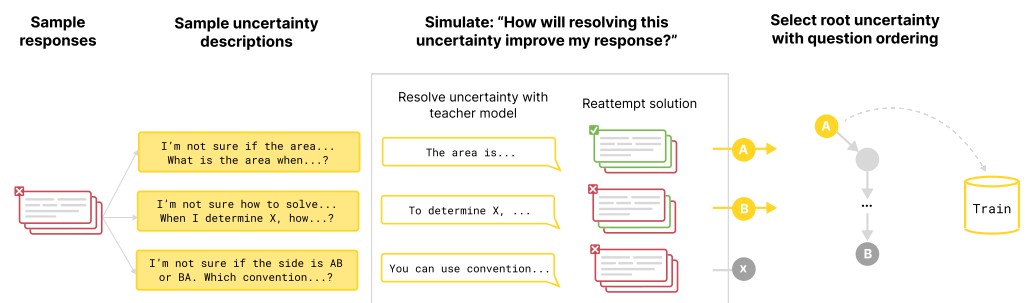

Figure 3: Overview of our method. First we sample $n$ chain-of-thought responses from the model $M$, and then we show all samples to $M$ and sample $k$ descriptions of its uncertainty. For each candidate uncertainty description, we ask a teacher model with access to the solution for the information that would resolve the uncertainty. We then prompt $M$ to edit each of its initial $n$ attempts given the teacher's response. We filter for the uncertainties which, when resolved, significantly improve $M$'s performance ("critical" uncertainty). Finally, we order each uncertainty and select the earliest ones to add to a training set ("root" uncertainty). The training set is used for supervised fine-tuning to reinforce $M$ generating diagnostic uncertainties.

Formally, we define an uncertainty at node $v$ as **diagnostic** if it is both:

**Critical**: Providing the value of $v$ sufficiently determines the downstream computation to solve the problem correctly

**Root**: No ancestor node $u$ of $v$ satisfies the critical property

In the next section, we propose a practical method to select for critical and root uncertainty.

We have described a setup where resolving uncertainty at a single node is sufficient for correctness, these definitions can be generalized to *sets* of uncertainties in instances where e.g. there are multiple challenging steps in a long-horizon task that have to be resolved one-by-one. Additionally, we limit our focus in this paper to resolving uncertainty makes the model's answers *more correct* under a ground-truth metric like exact match. More generally, critical nodes $v$ can be defined as those where fixing the value of $v$ *improves* the model's response, e.g. under a reward model.

## 4    METHOD

Our goal is to train a model $M$ to reliably output diagnostic uncertainty descriptions for any given question it is asked. Our graph-theoretic framework provides a precise definition of what kinds of descriptions we would like the model to output, but implementing it directly would require enumerating all possible solution paths and uncertainties. Instead, we approximate the search for diagnostic uncertainty descriptions by sampling from the LM itself.

We assume that we have a dataset $D$ of questions and answers to train model $M$ to output its uncertainty. We first sample $n = 20$ chain-of-thought attempts from $M$. Then we prompt $M$ to generate $k = 5$ uncertainty descriptions given the problem and its $n$ attempts by asking "Given this problem and your sampled responses, what are you most uncertain about?" Without training $M$ to output diagnostic uncertainty descriptions, we expect that some of the $k$ sampled descriptions will be spurious (e.g. unnecessarily hedging on a fact $M$ is already certain about), or no samples will identify $M$'s root uncertainty (and instead only identify uncertainties about downstream facts). To address this, we *filter* the sampled descriptions for those that are critical and root, train the model to output these diagnostic uncertainties, and then *iterate* this process several times to bootstrap more training data. On each iteration $i$, we use $M_{i-1}$, $M$ finetuned on all diagnostic descriptions that we discovered at the previous iterations $1, ..., i-1$, to generate $k = 5$ new candidates. This iterative process bootstraps the model's ability to generate better uncertainty descriptions. We describe this process in more detail below.

**Critical Uncertainty: Teacher Model**    First, out of the sampled uncertainty descriptions, we filter for critical uncertainty. To do so, we prompt $M$ to formulate its uncertainty into a question that would resolve its uncertainty, and then answer the question with a "teacher" `gpt-4o` model with access to the problem solution. For example, if the model is uncertain about interpreting "twice the difference," the teacher would use the solution to clarify "In this context, 'twice the difference' means you should first subtract the numbers, then multiply by 2." For each (uncertainty+query, answer) pair, we prompt $M$ to edit each of its $n$ original responses independently, given this new information after resolving the uncertainty it asked about. An uncertainty is considered critical if $M$ is able to improve its accuracy on the problem after editing. We evaluate whether the improvement is statistically significant with the exact sign test with significance threshold $\alpha = 0.05$.

**Root Uncertainty: Pairwise Comparison**    Out of the descriptions that are critical, we then identify which uncertainties are root. To do so, we train a pairwise ordering judge $J$ that determines for two uncertainties $A$ and $B$ which question is "upstream" – i.e., whether $A$ strictly needs to be answered first, $B$ strictly needs to be answered first, or neither, e.g. in the case where A and B are incomparable steps in different solution approaches. Our goal is to use the judge to select for examples where the model expresses more specific uncertainty when possible. For example:

```
Problem: For how many two-digit prime numbers is the sum of its digits 8?
A: How can I definitively determine if a number is prime or not?
B: What are the prime numbers among the two-digit numbers that can be formed
from digit pairs where the sum of the digits equals 8?
ANSWER: A
```

In the example above $A$ is upstream since a knowing how to determine if numbers are prime in general is necessary to find a set of prime numbers with a specific condition.

We train the judge on a small hand-labeled dataset to distinguish very specific queries from obvious "giveaway" queries that simply ask for the answer. For each of $d$ problems in the dataset, we generate $k' = 20$ possible uncertainty queries and identify queries that are either S=Specific or G=Giveaways. Some queries may be neither; we only label the clear ones. We then construct the judge dataset as follows:

- $\forall s_1, s_2 \in$ S $: ((s_1, s_2), -), ((s_2, s_1), -)$
- $\forall s \in$ S$, g \in$ G $: ((s, g), A), ((g, s), B)$
- $\forall g_1, g_2 \in$ G $: ((g_1, g_2), -), ((g_2, g_1), -)$

where a label of $A$ means the former query was upstream, $B$ means the latter, and $-$ means neither.

If there are any inconsistencies in the judge's output, e.g. if there is a cycle among a set of queries, we consider all those queries equally upstream. The judge model achieves 75% accuracy at this task on a held-out validation set.

Out of the critical uncertainty descriptions, we order all the descriptions with the judge to determine a set of questions that are upstream in the graph compared to the rest of the questions, and add this to the training set. This process approximates the search for root uncertainty by selecting for uncertainty that is most upstream out of the sample set.

**Iterative Training**    Finally, we train $M$ to output the uncertainty descriptions that are both critical and root, where each training example consists of:

```
<problem>
<n sampled attempts from M>
What are you most uncertain about?
< uncertainty: 'I'm unsure what the name of...'>
< query: 'What is the name of...?'>
```

We avoid reinforcing the tokens of the sampled attempts and only train $M$ with a loss on the tokens of the uncertainty description and query.

On each iteration, we add the selected uncertainties and queries to the training dataset and train from the base model (rather than the fine-tuned $M$ from the previous iteration). On the next iteration, we sample candidate uncertainty description from the finetuned model. We repeat this process for several iterations until the number of new examples plateaus. We choose to retrain from the base model on each iteration because we found that fine-tuning models from previous iterations was prone to overfitting.

At a high level, our method has similarities to hallucination detection methods that work by aggregating many samples (Farquhar et al., 2024), but in our case the model aggregates its own samples to determine an open-ended description of its uncertainty. By selecting for critical and root uncertainty, we ensure that these descriptions are actually important to solving the problem, rather than spurious or superficial differences between the samples (such as different ways to format the answer).

## 5 EXPERIMENTS

We run our experiments on Hendrycks MATH, using `gpt-4o-mini` as the base model that we train to express uncertainty.

We focus on investigating whether the model can generate good uncertainty descriptions on problems where it is incorrect, using a dataset of 1220 examples where `gpt-4o-mini`'s base empirical accuracy (out of 20 samples) is between 20% and 75% (885 train, 335 validation). Our goal is to produce open-ended uncertainty descriptions that are more meaningful to human users. However, to *automatically* evaluate whether these descriptions are more meaningful than those generated by zero-shot prompting, we measure several proxy metrics:

- **Edit Accuracy**: accurately identifying the part of the solution it's uncertain about should enable the model to *improve its response* after it resolves the uncertainty with the teacher model. We measure the model's accuracy after asking questions generated by our finetuned model, relative to questions generated by the baselines.

- **Entropy Reduction**: accurately identifying the part of the solution it's uncertain about should enable the model to *reduce the entropy in its answer distribution* after it resolves the uncertainty. We measure the difference between the model's answer distribution entropy before and after asking the question, relative to questions generated by the baselines.

We compare against three baselines, (1) `just ask (no samples)`, (2) `just ask (with samples)`, and (3) `edit without teacher`. For (1) and (2), we prompt the model to generate a query to the teacher and do not fine-tune, simply taking the out-of-the-box results. In (1), the model only sees the overall question; in (2) it is additionally given $n = 20$ of its own samples to generate a better query. For (3), we prompt the model for an uncertainty, allow it to edit its response given the uncertainty, but forgo the teacher response and the iterative bootstrapping.

For all experiments and methods, we evaluate the question generated by greedy decoding from the model. Following OpenAI's `simple-evals` evaluation harness, we grade answers by comparing the model response to the ground-truth answer with `gpt-4o-mini` to allow equivalent answers (grading 1.5 as correct if the true answer is $\frac{3}{2}$).

For our final model, we ran two iterations of training on successful questions to create model $M$, and compared its performance to four baselines. The results are in figure Figure 4. The model gets the highest accuracy when given samples to generate questions from, and our model gets 5% higher accuracy on validation after training.

### 5.1 ENTROPY REDUCTION

In order to analyze the entropy of the answer distribution, we merge sampled answers into groups of equivalent responses. Since equivalent responses may be presented in various different forms, a request is made to `gpt-4o-mini` determine if each pair of responses is equivalent or not, in the same manner that responses are graded. We compute the average cross-entropy, per problem, across the model's answers before and after editing, using log base 2. We then take the difference to measure entropy reduction.

Task Accuracy after Editing

Figure 4: The final accuracy of the model on problems on different settings. "Base Task Accuracy" is the model's accuracy on attempting the problems with no chance to edit. "Zero-shot, question only," "Zero-shot, 20 samples,", and "Self-Critique (no teacher answer)" correspond to baselines (1), (2), and (3), respectively. "Ours" refers to our fine-tuned model $M$.

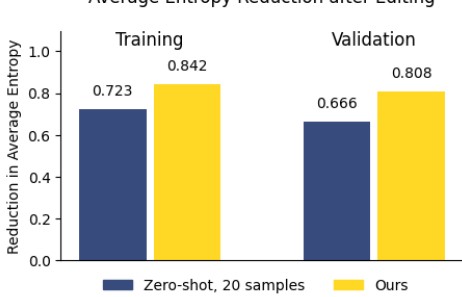

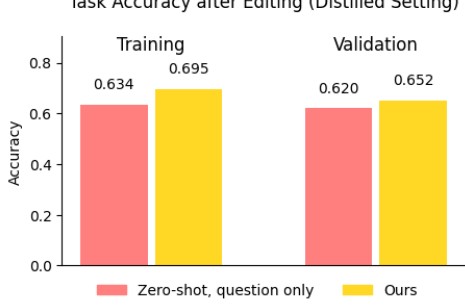

Figure 5: The difference in entropy before and after editing. "Zero-shot, 20 samples" baseline (2), and "Ours" is when model $M$ asks questions.

Figure 6: Accuracy on problems after editing. "Zero-shot, question only" is baseline (1). "Ours" is a our trained model $M'$.

As seen in figure Figure 5, our model model has about 21% more entropy reduction on held out validation compared to a model without training. Moreover, editing in general can greatly decrease entropy due to hints that make more consistent decisions.

## 5.2 DISTILLATION

In practice, generating 20 samples to determine uncertainty may be impractical in settings where the time to do so is significant. We investigated the setting of baseline (1) to see if the model is able to diagnose its uncertainty on problems without viewing its own samples. We will refer to this setting as the distilled setting.

To consider how fine-tuning can be used to improve questions in the distilled setting, we trained a model $M'$, and evaluated it in the distilled context. $M'$ is fine-tuned from the same model as $M$, gpt-4o-mini. To prepare it for the distilled context, the training examples for $M$ were modified to no longer have sample solutions, mimicking the distilled context. Then, gpt-4o-mini was fine-tuned with this set of modified examples to produce $M'$.

As seen in figure Figure 6, our model $M'$ achieved 3% higher accuracy relative to gpt-4o-mini. On the other hand, $M'$ had accuracy on training data similar to model $M$ (0.695 compared to 0.701), so the distilled setting may be particularly susceptible to over fitting.

## 5.3 QUALITATIVE RESULTS

```
Problem: Two sides of an isosceles triangle are 10 inches and 20 inches. If the
shortest side of a similar triangle is 50 inches, what is the perimeter of the larger
triangle?
```

```
gpt-4o-mini: One step where I was most uncertain was in determining the correct
configuration of the isosceles triangle given the sides of 10 inches and 20 inches.
I had to decide whether the two equal sides were 10 inches or 20 inches, and this
affected the validity of the triangle based on the triangle inequality theorem.
```
**How can I definitively determine the correct configuration of the isosceles triangle when given two side lengths?**

```
Model M: One step where I was most uncertain was in determining the correct
configuration of the isosceles triangle with the given side lengths of 10 inches
and 20 inches. I had to decide whether the two equal sides should be 10 inches or 20
inches, and I was unsure about the implications of the triangle inequality in each
case.
```
**How can I definitively determine which configuration of the triangle (10, 10, 20 or 20, 20, 10) is valid based on the triangle inequality theorem?**

In the example uncertainty descriptions above, model $M$ asks a question regarding a specific piece of information, while gpt-4o-mini asks for a more general approach. Model $M$'s question was successful in resolving uncertainty, with all samples being correct after editing. gpt-4o-mini's question, on the other hand, only resulting in one of seven incorrect samples being edited; for the other samples, the model saw the approach of using triangle inequality and assumed it had done so correctly.

## 6 DISCUSSION AND FUTURE WORK

**RL framing**    Identifying precise uncertainty in the general case requires using real-valued rewards to incentivize asking only what is necessary. In this work we used supervised learning instead of RL which required making the training data binary: we reinforced the critical uncertainties that are root *enough*, and filter out the rest. This is an approximation of the true desired objective. In future work, we would like to explore using a real-valued reward that penalizes the model proportional to the amount of information asked for. The optimal policy in this scheme would ask for precisely what information is needed and not more. This desired behavior is akin to information gathering in a POMDP (Partially Observable Markov Decision Process) Kaelbling et al. (1998).

**Effect of teacher model** For our method, it is important that the teacher answers the student's precise query. If the teacher model does not understand the query, it will not be able to provide a helpful response. This is a desired effect if the student's query was poorly phrased because it incentivizes the student to ask clear questions, but if the teacher model struggles to answer a clear query then our method may fail to identify diagnostic uncertainty. Conversely, sometimes the teacher model may try to provide more information than the student asked. This is much worse for our method because it may result in false positives: identifying an irrelevant query as the model's diagnostic uncertainty because it meaningfully improved the model's performance when in fact the teacher simply gave away the entire solution. This effect should be minimized for our method to be robust. In future work it would be interesting to explore different possible teachers, e.g. tools such as calculators or maps.

**Broader setting** We made several simplifying assumptions that could be relaxed in future work: (1) investigate continuous improvement on the overall problem rather than binarized pass/fail, and (2) extend the method to problems where the model has multiple uncertainties and/or may need multiple turns of interaction with a teacher to improve performance.

More broadly, it would be interesting to analyze model internals and explore whether and to what extent they explicitly represent uncertainty. It would also be interesting to consider categorical uncertainties, i.e. those that generalize across problems, for example if a model systematically lacks training data about a low-resource language.

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
