# OpenReview forum: "Diagnostic Uncertainty: Teaching Language Models to Describe Open-Ended Uncertainty"
_ICLR.cc/2025/Workshop/BuildingTrust — BuildingTrust_

### Official Review · Reviewer_bpCH · 2025-03-01
**Review of Submission Number 115**

**Rating:** 7
**Confidence:** 4

**Review:**

This paper introduces the notion of ‘diagnostic uncertainty’ - rather than an LLM providing a general uncertainty estimate of its answer, it is more valuable in many settings to obtain specifically the step(s) that it is critically uncertain about - that is, where clarification would have the biggest impact on downstream accuracy - and is also the root - the first such step.

To do this, the authors propose to fine-tune a model to verbalize their uncertainty about such a step, by first asking it to state which step it is most uncertain about over multiple samples, and then using a teacher model equipped with the correct answer to measure in which of those steps positive accuracy improvement is obtained by provision of the information. Furthermore a ‘judge model’ is trained to order these steps so that the root step can be identified. This fine-tuning-and-generation cycle is repeated multiple times.

The authors demonstrate that their method elicits the highest accuracy improvement and entropy reduction over baselines.

Overall, I encourage acceptance of this paper as I expect that it will foster significant discussion of interest amongst those working in or adjacent to this area, despite the paper’s weaknesses enumerated below.

## Strengths:

1. The paper discusses an interesting extension of an important area in current LLM research - uncertainty calibration/elicitation - into diagnostic uncertainty, which to my knowledge has not hitherto been examined extensively in the literature.
2. The paper proposes an interesting conceptual framework in which to understand diagnostic uncertainty.
3. The paper is relatively clear to follow - the motivations and baselines considered are reasonable.

## Weaknesses:

1. Although the motivation in Section 3.1 suggests that Criticality is defined as: ‘[clarifying the step’s uncertainty] determines the downstream computation to solve the problem correctly’ - this is not the measure used for criticality filtering in the method. Explicitly, the model is asked first to state ‘what [it is] most uncertain about’ which is not precisely the same thing. Moreover, the actual criticality filtering method takes all such queries which improve downstream performance - which is, again, not precisely the same thing - a more close match to the conceptual motivation would be taking all steps which result in 100% accuracy. In general, I encourage the authors to reflect more deeply on precisely what the desiderata are of the diagnostic uncertainty that they wish to extract, and then design a method that maps to that more faithfully; or, if not, to support why they deviate from that with experimental justification or otherwise.
2. Similarly, although the conceptual framework’s description of Rootness is clear, I am not convinced that this translates cleanly into real solutions. In real solutions, it may not be the case that there is a clear notion of ‘upstream’ or ‘downstream’ and these may also be very hard to determine - the analogy to traversal of a directed graph is, I think, only a very fuzzy one in reality. As the authors themselves also point out, rootness may be a feature of sets of nodes rather than a single node - which is the implicit assumption that is being made in the methodology and experiments presented. In practice, the method proposed also relies on a hand-labelled dataset, which is difficult to scale, especially to other more complex tasks. In my view, rootness - though an interesting property - is less important than criticality, and I would suggest that the authors focus most on diagnostic uncertainty specifically for maximising accuracy/performance first.
3. As the authors mention in Section 6, there is seemingly no filtering to ensure the teacher model is not providing significantly more information than the requested step’s clarification, jeopardizing the interpretation of the results.
4. For a future conference-level submission, the authors should ensure the results are replicated on a wider set of datasets/tasks, and ideally models as well.

## Questions:

1. For the judge - it is not clear if the 75% accuracy on the validation set is ‘good’. Is the baseline there 50% - is the score measured over pairwise accuracy, or is it the accuracy of the full ranking by repeated application of the pairwise judge?).
2. In lines 301-303, should the upstream labels A and B be the case for s1 and s2, rather than when there is a giveaway query g? If not - I do not understand why the model is not trained on two specific queries - that would seem to be the entire point of the judge?
3. How have you ensured that the MATH dataset/task fits into the framework of a single node correction being sufficient for correctness, rather than a set? If this is not ensured, or the case, then what is the repercussion of the assumption the method is making when applied to this problem?
4. Why do you retrain from the base model each time (lines 343-345) rather than iterate continually on the latest finetune? I did not spot a justification for this choice.

---

### Official Review · Reviewer_ck23 · 2025-03-02
**The authors proposed a novel way in an attempt to to identify the root cause of the response uncertainty of LLMs.**

**Rating:** 6
**Confidence:** 5

**Review:**

**Strengths**

- **Research Problem Identification:**
 The authors proposed a novel way in an attempt to to identify the root cause of the response uncertainty of LLMs. This can be more effective than the vanilla verbalized uncertainty.

- **Method:**
The authors proposed a practical method to fine-tune the model to improve its ability to output a more specific uncertain aspect of the task that is more crucial for reducing the response uncertainty and improve its quality.

- **Experiments/Results:**
  The author showed on a math dataset their more nuanced verbalized diagnostic uncertainty is more effective at pinpointing the cause of model's response uncertainty.

**Weaknesses**
- **Assumption on What Constitutes 'Uncertainty':**
The authors said 'Our key idea is that a model can be said to be uncertain about X if knowing X would improve its responses', but the authors also agree that 'the standard training methods for language models do not incentivize faithfulness by default'. So this is also possible that the model is blindly confident about a false belief, which it will not verbalize and result in false negatives.

- **Uncertainty Decomposition:**
  While the uncertainty that is both critical and root is important, the author may also want to consider other sources of uncertainty (technically every reasoning steps can have some degree of uncertainty). In particular, it is not always the case that addressing the most upstream uncertainty gives rise to the most uncertainty reduction, let alone guaranteeing the elimination of all downstream uncertainties.


- **Objective of the Judge Model J:**
       The authors stated that 'Our goal is to use the judge to select for examples where the model expresses more specific uncertainty when possible', but the end goal is to select the more 'upstream' uncertainty. However, 'more specific' does not necessitate 'more upstream' in many cases. Think of a tree search problem with larger branching factor near the leaf node.

- **Root Uncertainty Selection Criteria in the Experiment vs in the Definition:**
  In practice, it is very challenging to select the root uncertainty strictly according to its counter-factual definition given in line 080, as it requires to check if after addressing the candidate of root uncertainty, address the all the rest of the uncertainties will not lead to improvement for model response. In practice, it can be the case that even after eliminating the $k$ most upstream uncertainties, there will still be residual uncertainty left in the model response. Therefore, there is some degree of inconsistency in how root uncertainty is defined vs how it is checked in the experiment. Unless making this clear, the authors are conflating their own definition of root uncertainty and the notion of upstream uncertainty.

- **Lack of Results on Failure Cases Due to Teacher Model:**
In Line 468-475 the authors clearly listed down the cases where the teacher model fails to provide needed feedback to the student model, but in the paper the relevant results were not shown.

**Question**

- L344: It is not clear to me why the author chose to 'add the selected uncertainties and queries to the training dataset and train from the base model (rather than the fine-tuned M from the previous iteration)'.

- L430: It is said 'As seen in figure Figure 6, our model M ′ achieved 5% higher accuracy relative to gpt-4o-mini', but I only saw around 3% improvement on the validation set. I might have misunderstood something here, but I think this part is not written clearly.

---

### Official Review · Reviewer_UBYp · 2025-03-02
**Good contribution in using uncertainty to solve**

**Rating:** 7
**Confidence:** 2

**Review:**

This is a good contribution that demonstrates that training models to identify which steps of solving a problem they are most uncertain about improves overall task accuracy. I would encourage the authors to extend to other model families and more different task types.

---

### Decision · Program_Chairs · 2025-03-04

Accept